# Somatic mutation rates scale with time not growth rate in long-lived tropical trees

Akiko Satake[1]*[†], Ryosuke Imai[1†], Takeshi Fujino[2], Sou Tomimoto[1], Kayoko Ohta[1], Mohammad Na'iem[3], Sapto Indrioko[3], Widiyatno Widiyatno[3], Susilo Purnomo[4], Almudena Molla Morales[5], Viktoria Nizhynska[5], Naoki Tani[6,7], Yoshihisa Suyama[8], Eriko Sasaki[1], Masahiro Kasahara[2]

[1]Department of Biology, Faculty of Science, Kyushu University, Fukuoka, Japan; [2]Department of Computational Biology and Medical Sciences, Graduate School of Frontier Sciences, The University of Tokyo, Chiba, Japan; [3]Faculty of Forestry, Universitas Gadjah Mada, Sleman, Indonesia; [4]PT. Sari Bumi Kusuma, Pontianak, Indonesia; [5]Gregor Mendel Institute of Molecular Plant Biology, Austrian Academy of Sciences, Vienna, Austria; [6]Forestry Division, Japan International Research Center for Agricultural Sciences, Tsukuba, Japan; [7]Faculty of Life and Environmental Sciences, University of Tsukuba, Tsukuba, Japan; [8]Field Science Center, Graduate School of Agricultural Science, Tohoku University, Osaki, Japan

*For correspondence:
akiko.satake@kyudai.jp

[†]These authors contributed equally to this work

Competing interest: The authors declare that no competing interests exist.

## eLife assessment

Satake and colleagues' **important** study elucidates somatic mutation processes in plants, demonstrating that in two tropical trees, mutation rates correlate with age, not growth rates. Their **convincing** evidence shows that many mutations do not align with cell divisions, suggesting many somatic mutations are generated in a replication-independent manner. This study represents a significant step towards advancing our understanding of plant development and the patterns and inheritance of mutations. This significant research is poised to engage a diverse array of scholars in plant evolution and development.

**Abstract** The rates of appearance of new mutations play a central role in evolution. However, mutational processes in natural environments and their relationship with growth rates are largely unknown, particular in tropical ecosystems with high biodiversity. Here, we examined the somatic mutation landscapes of two tropical trees, *Shorea laevis* (slow-growing) and *S. leprosula* (fast-growing), in central Borneo, Indonesia. Using newly constructed genomes, we identified a greater number of somatic mutations in tropical trees than in temperate trees. In both species, we observed a linear increase in the number of somatic mutations with physical distance between branches. However, we found that the rate of somatic mutation accumulation per meter of growth was 3.7-fold higher in *S. laevis* than in *S. leprosula*. This difference in the somatic mutation rate was scaled with the slower growth rate of *S. laevis* compared to *S. leprosula*, resulting in a constant somatic mutation rate per year between the two species. We also found that somatic mutations are neutral within an individual, but those mutations transmitted to the next generation are subject to purifying selection. These findings suggest that somatic mutations accumulate with absolute time and older trees have a greater contribution towards generating genetic variation.

## Introduction

Biodiversity ultimately results from mutations that provide genetic variation for organisms to adapt to their environments. However, how and when mutations occur in natural environments is poorly understood (*Whitham and Slobodchikoff, 1981*; *Gill et al., 1995*; *Schoen and Schultz, 2019*). Recent genomic data from long-lived multicellular species have begun to uncover the somatic genetic variation and the rate of naturally occurring mutations (*Yu et al., 2020*; *Reusch et al., 2021*). The rate of somatic mutations per year in a 234-year-old oak tree has been found to be surprisingly low (*Schmid-Siegert et al., 2017*) compared to the rate in an annual her (*Ossowski et al., 2010*). Similar analyses in other long-lived trees have also shown low mutation rates in both broadleaf trees (*Plomion et al., 2018*; *Wang et al., 2019*; *Orr et al., 2020*; *Hofmeister et al., 2020*; *Duan et al., 2022*) and conifers (*Hanlon et al., 2019*). Despite the growing body of knowledge of somatic mutation landscapes in temperate regions, there is currently no knowledge on the somatic mutation landscapes in organisms living in tropical ecosystems, which are among the most diverse biomes on Earth.

Mutations can arise from errors during replication (*Reijns et al., 2015*), or from DNA damage caused by exogenous mutagens or endogenous reactions at any time during cell growth (*Gao et al., 2016*). While DNA replication errors have long been assumed to be major sources of mutations (*Makova and Li, 2002*; *Tomasetti and Vogelstein, 2015*), a modeling study that relates the mutation rate to rates of DNA damage, repair and cell division (*Gao et al., 2016*) and experimental studies in yeast (*Liu and Zhang, 2019*), human (*Abascal et al., 2021*), and other animals *de Manuel et al., 2022* have shown the importance of mutagenic processes that do not depend on cell division. Consequently, it remains largely unknown which source of mutations, whether replicative or non-replicative, predominates in naturally growing organisms.

To investigate the rates and patterns of somatic mutation and their relation to growth rates in tropical organisms, we studied the somatic mutation landscapes of slow- and fast-growing tropical trees in a humid tropical rain forest of Southeast Asia. By comparing the somatic mutation landscape between slow- and fast-growing species in a tropical ecosystem, we can gain insights into the mutagenesis that occurs in a natural setting. This comparison provides a unique opportunity to understand the impact of growth rate on somatic mutations and its potential role in driving evolutionary processes.

## Results

### Detecting somatic mutations in slow- and fast-growing tropical trees

The humid tropical rainforests of Southeast Asia are characterized by a preponderance of trees of the Dipterocarpaceae family (*Ghazoul, 2016*). Dipterocarp trees are highly valued for both their contribution to forest diversity and their use in timber production. For the purposes of this study, we selected *Shorea laevis* and *S. leprosula*, both native hardwood species of the Dipterocarpaceae family (*Figure 1—figure supplement 1a*). *S. laevis* is a slow-growing species (*Widiyatno et al., 2014*), with a mean annual increment (MAI) of diameter at breast height (DBH) of 0.38 cm/year (as measured over a 20-year period in n=2 individuals; *Supplementary file 1a*). In contrast, *S. leprosula* exhibits a faster growth rate (*Widiyatno et al., 2014*), with an MAI of 1.21 cm/year (n=18; *Supplementary file 1a*), which is 3.2 times greater than that of *S. laevis*. We selected the two largest individuals of each species (S1 and S2 for *S.laevis* and F1 and F2 for *S. leprosula*; *Figure 1a*) at the study site, located just below the equator in central Borneo, Indonesia (*Figure 1—figure supplement 1b*). We collected leaves from the apices of seven branches and a cambium from the base of the stem from each tree (*Figure 1a*; *Figure 1—figure supplement 2*), resulting in a total of 32 samples. To determine the physical distance between the sampling positions, we measured the length of each branch (*Supplementary file 1b*) and DBH (*Supplementary file 1c*). The average heights of the slow- and fast-growing species were 44.1 m and 43.9 m, respectively (*Figure 1a*; *Supplementary file 1c*). While it is challenging to accurately estimate the age of tropical trees due to the absence of annual rings, we used the DBH/MAI to approximate the average age of the slow-growing species to be 256 years and the fast-growing species to be 66 years (*Supplementary file 1c*).

To identify somatic mutations, we constructed new reference genomes of the slow- and fast-growing species. We generated sequence data using long-read PacBio RS II and short-read Illumina sequencing and assembled the genome using DNA extracted from the apical leaf at branch 1–1 of the tallest individual of each species (S1 and F1). The genomes were estimated to contain 52,935

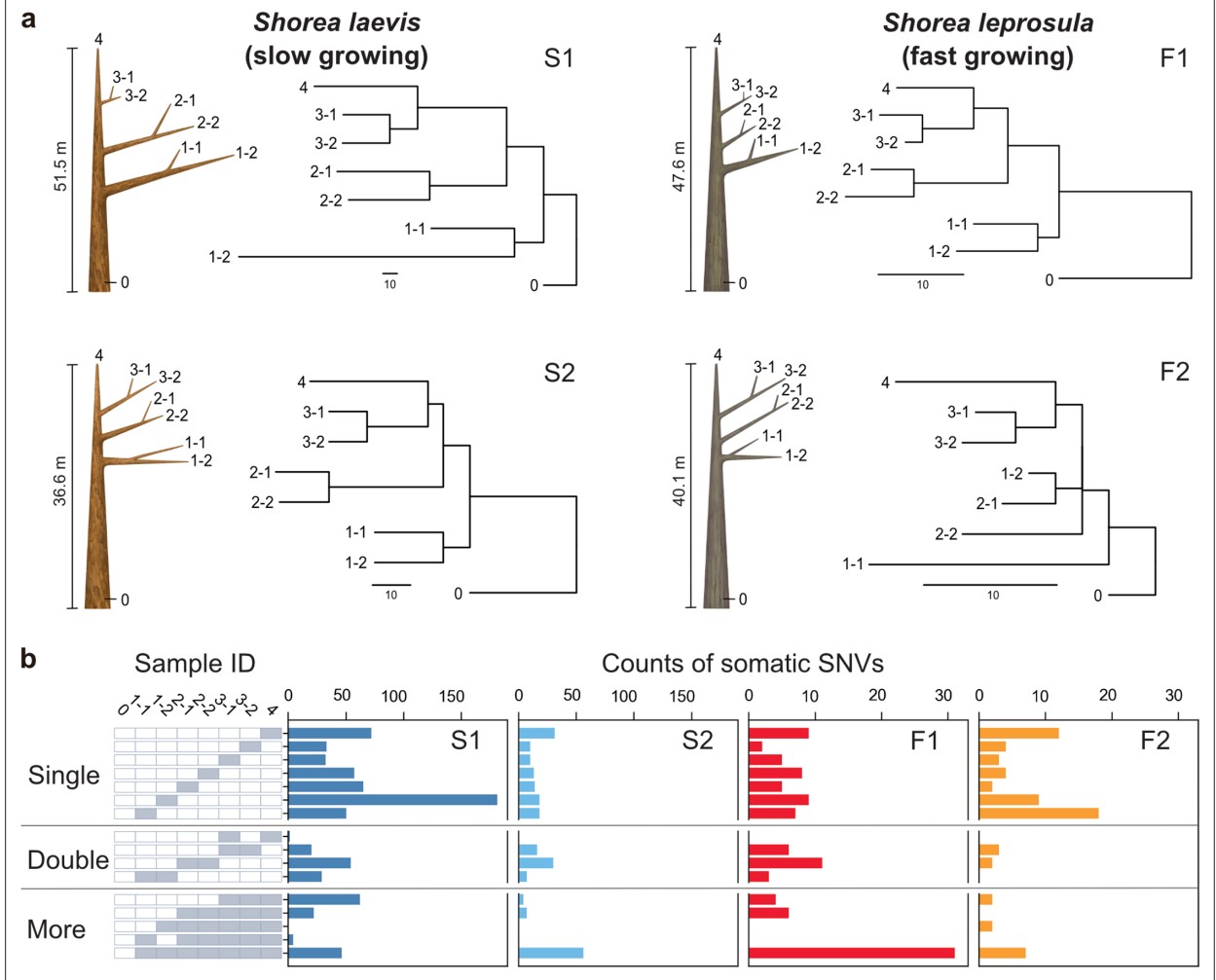

**Figure 1.** Physical tree structures and phylogenetic trees constructed from somatic mutations. (**a**) Comparisons of physical tree structures (left, branch length in meters) and neighbor-joining (NJ) trees (right, branch length in the number of nucleotide substitutions) in two tropical tree species: *S. laevis*, a slow-growing species (S1 and S2), and *S. leprosula*, a fast-growing species (F1 and F2). IDs are assigned to each sample from which genome sequencing data were generated. Vertical lines represent tree heights. (**b**) Distribution of somatic mutations within tree architecture. A white and gray panel indicates the presence (gray) and absence (white) of somatic mutation in each of eight samples compared to the genotype of sample 0. Sample IDs are the same between panels (**a**) and (**b**). The distribution pattern of somatic mutations is categorized as Single, Double, and More depending on the number of samples possessing the focal somatic mutations. Among $2^7 - 1$ possible distribution patterns, the patterns observed in at least one of the four individuals are shown.

The online version of this article includes the following figure supplement(s) for figure 1:

**Figure supplement 1.** Target tropical trees and location of study site.

**Figure supplement 2.** Workflow for identifying de novo somatic SNVs.

**Figure supplement 3.** Synteny relationship between *S. laevis* and *S. leprosula*.

and 40,665 protein-coding genes, covering 97.9% and 97.8% of complete BUSCO genes (eudicots_odb10) for the slow- and fast-growing species (*Supplementary file 1d*). Genome sizes estimated using k-mer distribution were 347 and 376 Mb for the slow- and fast-growing species, respectively. The synteny relationship between *S. laevis* and *S. leprosula* exhibited a high level of conservation overall (*Figure 1—figure supplement 3*).

To accurately identify somatic mutations, we extracted DNA from each sample twice to generate two biological replicates (*Figure 1—figure supplement 2*). A total of 64 DNA samples were sequenced, yielding an average coverage of 69.3 and 56.5×per sample for the slow- and fast-growing species, respectively (*Supplementary file 1e*). We identified Single Nucleotide Variants (SNVs) within the same individual by identifying those that were identical within two biological replicates of each sample

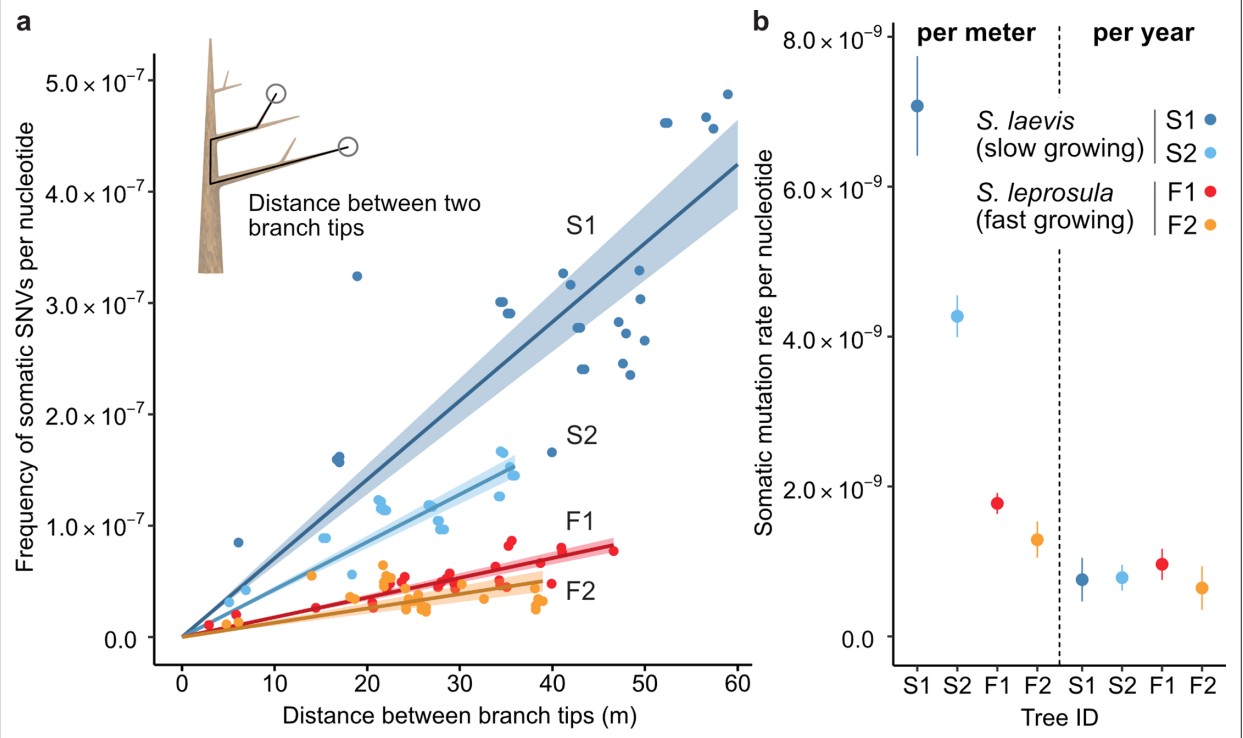

**Figure 2.** The relationship between the physical distance and the numbers of SNVs. (**a**) Linear regression of the number of SNVs against the pair-wise distance between branch tipcs with an intercept of 0 for each tree (S1: blue, S2: right blue, F1: red, and F2: orange). Shaded areas represent 95% confidence intervals of regression lines. Regression coefficients are listed in **Supplementary file 1h**. (**b**) Comparison of somatic mutation rates per nucleotide per growth and per year across four tropical trees. Bars indicate 95% confidence intervals.

(**Figure 1—figure supplement 2**). We identified 728 and 234 SNVs in S1 and S2, and 106 and 68 SNVs in F1 and F2, respectively (**Figure 1—figure supplement 2**; **Supplementary file 1f**). All somatic mutations were unique and did not overlap between individuals. We conducted an independent evaluation of a subset of the inferred single nucleotide variants (SNVs) using amplicon sequencing. Our analysis demonstrated accurate annotation for 31 out of 33 mutations (94% overall), with 22 out of 24 mutations on S1 and all 9 mutations on S2 (**Supplementary file 1g**).

## Somatic mutation rates per year is independent of growth rate

Phylogenetic trees constructed using somatic mutations were almost perfectly congruent with the physical tree structures (**Figure 1a**), even though we did not incorporate knowledge of the branching topology of the tree in the SNV discovery process. The majority of somatic mutations were present at a single branch, but we also identified somatic mutations present in multiple branches (**Figure 1b**) which are likely transmitted to new branches during growth. We also observed somatic mutations that did not conform to the branching topology (**Figure 1b**), as theoretically predicted due to the stochastic loss of somatic mutations during branching (**Tomimoto and Satake, 2023**).

Our analysis revealed that the number of SNVs increases linearly as the physical distance between branch tips increases (**Figure 2a**). The somatic mutation rate per site per meter was determined by dividing the slope of the linear regression of the number of SNVs against the physical distance between branch tips by the number of callable sites from the diploid genome of each tree (**Figure 2b**; **Supplementary file 1h**). The somatic mutation rate per nucleotide per meter was $7.08\times10^{-9}$ (95% CI: $6.41$–$7.74\times10^{-9}$) and $4.27\times10^{-9}$ (95% CI: $3.99$–$4.55\times10^{-9}$) for S1 and S2, and $1.77\times10^{-9}$ (95% CI: $1.64$–$1.91\times10^{-9}$) and $1.29\times10^{-9}$ (95% CI: $1.05$–$1.53\times10^{-9}$) for F1 and F2, respectively. The average rate of somatic mutation for the slow-growing species was $5.67\times10^{-9}$ nucleotide$^{-1}$ m$^{-1}$, which is 3.7-fold higher than the average rate of $1.53\times10^{-9}$ nucleotide$^{-1}$ m$^{-1}$ observed in the fast-growing species (**Figure 2b**; **Supplementary file 1h**). This result indicates that the slow-growing tree accumulates more somatic mutations compared to the fast-growing tree to grow the unit length. This cannot be explained by

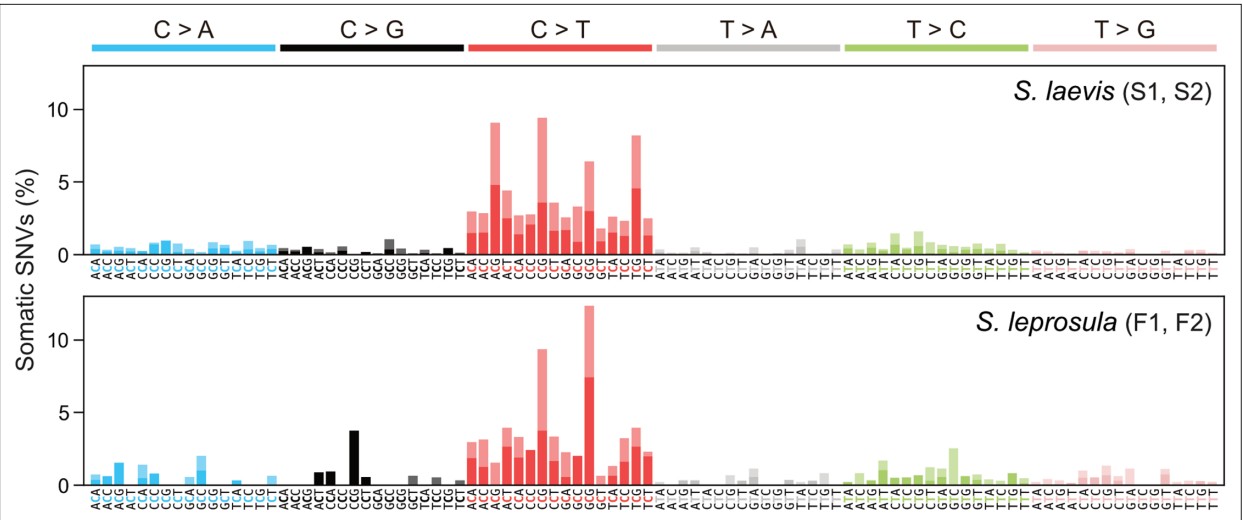

**Figure 3.** Mutational spectra of somatic SNVs. Somatic mutation spectra in *S. laevis* (upper panel) and *S. leprosula* (lower panel). The horizontal axis shows 96 mutation types on a trinucleotide context, coloured by base substitution type. Different colours within each bar indicate complementary bases. For each species, the data from two trees (S1 and S2 for *S. laevis* and F1 and F2 for *S. leprosula*) were pooled to calculate the fraction of each mutated triplet.

The online version of this article includes the following figure supplement(s) for figure 3:

**Figure supplement 1.** Mutational spectra of somatic and inter-individual substitutions.

**Figure supplement 2.** Manual confirmation of candidate SNVs.

**Figure supplement 3.** Proportion of potential false positive SNVs for *S. laevis* (S1, S2) and *S. leprosula* (F1, F2).

**Figure supplement 4.** Proportion of potential false negative SNVs for *S. laevis* (S1, S2) and *S. leprosula* (F1, F2).

differences in the number of cell divisions, as the length and diameter of fiber cells in both species are not substantially different 1.29 mm and 19.0 µm for the slow-growing species (*Usami, 1978*) and 0.91 mm and 22.7 µm for the fast-growing species (*Praptoyo and Mayaningsih, 2012*).

Based on the estimated age of each tree, somatic mutation rate per nucleotide per year was calculated for each tree. On average, resultant values were largely similar between the two species, with $7.71 \times 10^{-10}$ and $8.05 \times 10^{-10}$ nucleotide$^{-1}$ year$^{-1}$ for the slow- and fast-growing species, respectively (*Figure 2b*; *Supplementary file 1h*). This result suggests that somatic mutation accumulates in a clock-like manner as they age regardless of tree growth. The result suggests that somatic mutation accumulates in a clock-like manner as they age regardless of tree growth. Our estimates of somatic mutation rates per nucleotide per year in *Shorea* are higher than those previously reported in other long-lived trees such as *Quercus robur* (*Schmid-Siegert et al., 2017*), *Populus trichocarpa* (*Hofmeister et al., 2020*), *Eucalyptus melliodora* (*Orr et al., 2020*), and *Picea sitchensis* (*Hanlon et al., 2019*). This might suggest that long-lived trees in the tropics do not necessarily suppress somatic mutation rates to the same extent as their temperate counterparts. To validate this assertion, additional studies are required to compare somatic mutation rates among trees in tropical, temperate, and boreal regions, employing standardized methodologies.

## Mutational spectra are similar between slow- and fast-growing trees

Somatic mutations may be caused by exogenous factors such as ultraviolet and ionizing radiation, or endogenous factors such as oxidative respiration and errors in DNA replication. To identify characteristic mutational signatures caused by different mutagenic factors, we characterized mutational spectra by calculating the relative frequency of mutations at the 96 triplets defined by the mutated base and its flanking 5' and 3' bases (*Figure 3*; *Figure 3—figure supplement 1*). Across species, the mutational spectra showed a dominance of cytosine-to-thymine (C>T and G>A on the other strand, noted as C:G>T:A) substitutions at CpG sites with CG (*Figure 3a and b*). This is believed to result from the spontaneous deamination of 5-methylcytosine (*Coulondre et al., 1978*; *Duncan and Miller, 1980*). Methylated CpG sites spontaneously deaminate, leading to TpG sites and increasing the number of

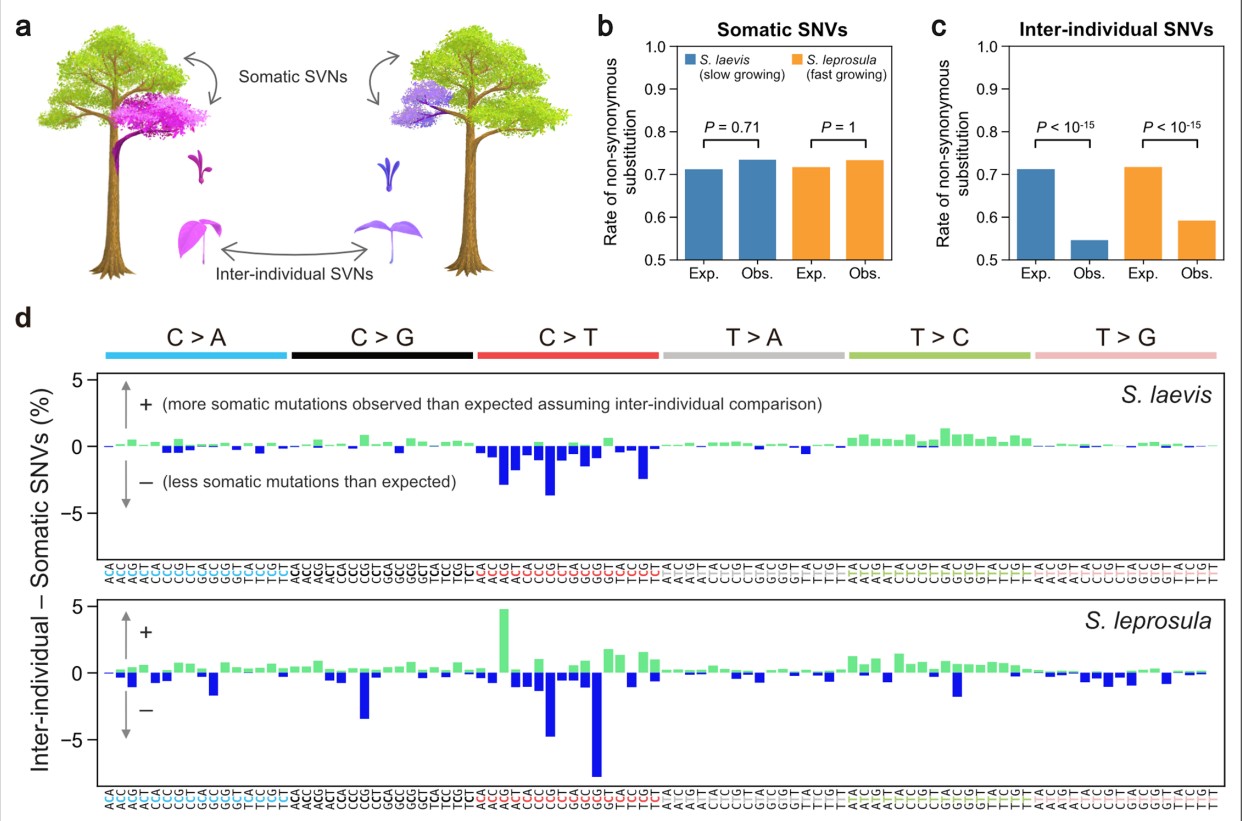

**Figure 4.** Detecting selection on somatic and inter-individual SNVs. (**a**) An illustration of somatic and inter-individual SNVs. Different colours indicate different genotypes. (**b**) Expected (Exp.) and observed (Obs.) rates of somatic non-synonymous substitutions. (**c**) Expected (Exp.) and observed (Obs.) rates of inter-individual non-synonymous substitutions. (**d**) The difference between the fractions of inter-individual and somatic substitutions spectra in *S. laevis* (upper panel) and *S. leprosula* (lower panel). The positive and negative values are plotted in different colours. The horizontal axis shows 96 mutation types on a trinucleotide context, coloured by base substitution type.

The online version of this article includes the following figure supplement(s) for figure 4:

**Figure supplement 1.** A calculation scheme for the expected rate of non-synonymous mutation.

C>T substitutions (*Cooper and Krawczak, 1989*). Compared to the proportion of CpG sites in the reference genomes, the proportion of somatic mutations at CpG sites showed a 3.38-fold and 2.56-fold increase for F1 and F2, and a 4.54-fold and 3.53-fold increase for S1 and S2, respectively.

We compared the mutational spectra of our tropical trees to single-base substitution (SBS) signatures in human cancers using the Catalogue Of Somatic Mutations In Cancer (COSMIC) compendium of mutation signatures (COSMICv.2 *Alexandrov et al., 2013*; *Nik-Zainal et al., 2016*; *Alexandrov et al., 2020*). The mutational spectra were largely similar to the dominant mutation signature in humans known as SBS1 (cosine similarity = 0.789 and 0.597 for the slow- and fast-growing species; *Supplementary file 1i*). SBS1 is believed to result from the spontaneous deamination of 5-methylcytosine. The mutational spectra were also comparable to another dominant signature in all human cancers, SBS5 (cosine similarity = 0.577 and 0.558 for the slow- and fast-growing species; *Supplementary file 1i*), the origin of which remains unknown. Our finding that somatic mutations in tropical trees accumulate in a clock-like manner (*Figure 2a*) is consistent with the clock-like mutational process observed in SBS1 and SBS5 in human somatic cells (*Alexandrov et al., 2015*; *Lee-Six et al., 2019*). This suggests that the mutational processes in plants and animals are conserved, despite the variation in their life forms and environmental conditions.

## Somatic mutations are neutral but inter-individual SNVs are subject to selection

We tested whether the somatic mutations and inter-individual SNVs are subject to selection (*Figure 4a*). The observed rate of non-synonymous somatic mutations did not deviate significantly from the expected rate under the null hypothesis of neutral selection in both the slow- (binomial test: p=0.71) and fast-growing (binomial test: p=1.0) species (*Figure 4b*; *Supplementary file 1j*). In contrast, the number of inter-individual SNVs were significantly smaller than expected (p<10$^{-15}$ for both species: *Figure 4c*). These results indicate that somatic mutations are largely neutral within an individual, but mutations passed to next generation are subject to strong purifying selection during the process of embryogenesis, seed germination, and growth.

Overall, the mutational spectra were similar between somatic and inter-individual SNVs (*Figure 3—figure supplement 1*). However, the fraction of C>T substitutions, in particular at CpG sites, was lower in inter-individual SNVs compared to somatic SNVs (*Figure 4d*). This observation may be indicative of the potential influence of GC-biased gene conversion during meiosis (*Duret and Galtier, 2009*) or biased purifying selection for C>T inter-individual nucleotide substitutions.

## Discussion

Our study demonstrates that while the somatic mutation rate per meter is higher in the slow- than in fast-growing species, the somatic mutation rate per year is independent of growth rate. To gain deeper understanding of these findings, we developed a simple model that decomposes the mutation rate per site per cell division (μ into the two components: DNA replication dependent ($\alpha$) and replication independent ($\beta$) mutagenesis). This can be represented as $\mu = \alpha + \beta\tau$, where $\tau$ is the duration of cell cycle measured in years. The replication dependent mutation emanates from errors that occur during DNA replication, such as the misincorporation of a nucleotide during DNA synthesis. The replication independent mutation arises from DNA damage caused by endogenous reactions or exogenous mutagens at any time of cell cycle. Since the number of cell division per year is given as $r = 1/\tau$, the mutation rate per year becomes $r\mu = \alpha/\tau + \beta$. From the relationship, the number of nucleotide substitution per site accumulated over *t* years, denoted as *m(t)*, is given by $m\left(t\right) = \left(\alpha/\tau + \beta\right)t$. The formula indicates that when $\beta$ is significantly greater than $\alpha$, somatic mutations accumulate with tree age rather than with tree growth.

We estimated the relative magnitudes of $\alpha$ and $\beta$ by using the results obtained from our study. Given that the cell cycle duration is likely inversely proportional to MAI, we have $\tau_S/\tau_F$ = 3.2 (*Supplementary file 1a*), where $\tau_S$ and $\tau_F$ denote the cell cycle duration for the slow- and fast-growing species, respectively. It is also reasonable to assume that the same number of cell divisions are required to achieve 1 m of growth in both species as the cell size is similar between the two species. Based on our estimates of the somatic mutation rate per site per meter for the slow- ($\mu_S$) and fast-growing species ($\mu_F$), we have $\mu_S/\mu_F = \left(\alpha + \beta\tau_S\right)/\left(\alpha + \beta\tau_F\right)$ = 3.7, which is close to the ratio of cell cycle duration $\tau_S/\tau_F$. This consistency can be explained by the substantial contribution of the replication independent mutagenesis to the somatic mutation rate (i.e. $\beta \gg \alpha$), as long as the magnitudes for $\alpha$ and $\beta$ are similar between the two species. The time required for a unit length to grow can vary even within the same species, depending on microenvironmental conditions such as the availability of light and nutrients. These variations could explain the differences in somatic mutation rates per unit growth between two individuals within the same species (*Figure 2*).

This argument concords with previous studies in human and other animals, which showed the presence of mutations that do not track cell division (*Abascal et al., 2021*; *de Manuel et al., 2022*). This study contributes to understanding the importance of non-replicative mutagenesis in naturally grown trees by decoupling the impacts of growth and time on the rate of somatic mutation. The preponderance of non-replicative mutational process can be attributed to its distinct molecular origin, the accumulation of spontaneous CpG mutations with absolute time. The neutral nature of newly arising somatic mutations within the tree results in a molecular clock, a constant rate of molecular evolution (*Zuckerkandl and Pauling, 1965*; *Kimura and Ota, 1971*; *Kimura, 1983*). For our argument, we made an intuitive assumption that the number of stem cell divisions increases with distance regardless of species when cell size is similar. However, to further validate this assumption, we require mathematical models that consider the asymmetric division of stem cells within the meristem (*Watson et al.,*

*2016*; *Lanfear, 2018*) and complex stem cell population dynamics during elongation and branching in tree growth (*Tomimoto and Satake, 2023*; *Iwasa et al., 2023*). Moreover, understanding establishment timing of germlines during development is crucial in addressing the impact of somatic mutation on the next generation (*Lanfear, 2018*). The model we have presented here is based on the assumption that genetic drift is prominent within a stem cell population, and that a single stem cell lineage becomes fixed within a meristem. However, future studies could explore relaxing this assumption to consider the contribution of multiple stem cell lineages. By doing so, we can gain insights into how the relationship between pairwise genetic differences and the distance between branch tips is influenced by the branching architecture of the tree and the strength of genetic drift. Furthermore, improving the accuracy of our argument, as derived from the model, can be achieved through future investigations that directly estimate the cell cycle duration for each individual tree.

The relative importance of replication independent mutagenesis, represented as the relative magnitude of $\beta$ compared to $\alpha$, can vary through evolution possibly through selection on DNA repair pathways. The selection pressure that leads to different magnitudes either or both for $\alpha$ or $\beta$ may explain the differential somatic mutation rate per year in mammals with different lifespan (*Cagan et al., 2022*). Conversely, in plants, the selection pressure to constrain somatic mutation rates to lower levels in long-lived trees might be less significant. A definitive answer to this query awaits the accumulation of additional data on somatic mutation rates in closely related plant species inhabiting the same environment but exhibiting different growth rates.

## Materials and methods
### Study site and sampling methods
The study site is in a humid tropical rain forest in Central Borneo, Indonesia (00°49′ 45.7″ S, 112°00′ 09.5″ E; *Figure 1—figure supplement 1b*). The forest is characterized by a prevalence of trees of the Dipterocarpaceae family and is managed through a combination of selective logging and line planting (Tebang Pilih Tanam Jalur, TPTJ). The mean annual temperature range from 2001–2009 was between 22 to 28°C at night and 30 to 33°C during the day, with an average annual precipitation of 3376 mm[41].

The study focuses on two native Dipterocarpaceae species, *S. laevis* and *S. leprosula* (*Figure 1—figure supplement 1a*). We logged two individuals from each species (S1 and S2 for *S. laevis* and F1 and F2 for *S leprosula*; *Figure 1—figure supplement 1a*) on July 17–18, 2018 and collected samples prior to their transportation for timber production. Approximately 0.4–1.0 g of leaf tissue was collected from each of the apices of seven branches and approximately 5 g of cambium tissue was taken from the base of the stem per individual (*Figure 1—figure supplement 2*). To calculate the physical distance between sampling positions within the tree architecture, we measured the length of each branch (*Supplementary file 1b*). Samples were promptly preserved in a plastic bag with silica gel following harvest and transported to the laboratory within 4 days of sampling. During transportation, samples were kept in a cooler box with ice to maintain a low temperature. Once in the laboratory, samples were stored at −80 °C until DNA and RNA extraction.

DBH have been recorded for the trees with DBH greater than 10 cm every two years since 1998 within three census plots of 1 hectare (100×100 m) in size located near the target trees. The mean growth was calculated by taking the average of MAI of DBH for 2 and 18 trees for the slow- and fast-growing species, respectively (*Supplementary file 1a*).

### DNA extraction
For short-read sequencing, DNA extraction was performed using a modified version of the method described previously (*Doyle and Doyle, 1987*) as follows: Frozen leaves were ground in liquid nitrogen and washed up to five times with 1 mL buffer (including 100 mM HEPES pH 8.0, 1% PVP, 50 mM Ascorbic acid, 2% (v/v) β-mercaptoethanol) (*Toyama et al., 2015*). DNA was treated with Ribonuclease (Nippongene, Tokyo, Japan) according to the manufacture's instruction. DNA was extracted twice independently from each sample for two biological replicates. The DNA yield was measured on a NanoDrop ND-2000 spectrophotometer (Thermo Fisher Scientific, Waltham, MA, USA) and Qubit4 Fluorometer (Thermo Fisher Scientific). For long-read sequencing, we extracted high-molecular-weight genomic DNA from branch 1–1 leaf materials of S1 and F1 individuals using a modified CTAB method (*Doyle, 1991*).

## RNA extraction and sequencing

For genome annotation, total RNA was extracted from the cambium sample of the S1 individual of *S. laevis* in accordance with the method described in a previous study (*Yeoh et al., 2017*). RNA integrity was measured using the Agilent RNA 6000 Nano kit on a 2100 Bioanalyzer (Agilent Technologies, Santa Clara, CA, USA), and the RNA yield was determined using a NanoDrop ND-2000 spectrophotometer (Thermo Fisher Scientific). The extracted RNA was sent to Pacific Alliance Lab (Singapore), where a cDNA library was prepared with a NEBNext Ultra RNA Library Prep Kit for Illumina (New England BioLabs, Ipswich, MA, USA) and 150 paired-end transcriptome sequencing was conducted using an Illumina NovaSeq6000 sequencer (Illumina, San Diego, CA, USA). For *S. leprosula*, we used published RNA-seq data (*Ng et al., 2021*).

## Illumina short-read sequencing and library preparation

For Illumina short-read sequencing, the DNA sample from the first replicate of the S1 individual of *S. laevis* was sent to the Next Generation Sequencing Facility at Vienna BioCenter Core Facilities (VBCF), a member of the Vienna BioCenter (VBC) in Austria, for library preparation and sequencing on the Illumina HiSeq2500 platform (Illumina). The library was prepared using the on-bead tagmentation library prep method according to the manufacturer's protocol and was individually indexed with the Nextera index Kit (Illumina) by PCR. Insert size was adjusted to around 450 bp. The quantity and quality of each amplified library were analyzed using the Fragment Analyzer (Agilent Technologies) and the HS NGS Fragment Kit (Agilent Technologies).

The DNA sample from the second replicate of the S1 individual and two replicates from the S2, F1, and F2 individuals were sent to Macrogen Inc (Republic of Korea) for sequencing on the Illumina HiseqX platform (Illumina). DNA was sheared to around 500 bp fragments in size using dsDNA fragmentase (New England BioLabs). Library preparation was performed using the NEBNext Ultra II DNA Library Prep Kit (New England BioLabs) according to the manufacturer's protocol, and the libraries were individually indexed with the NEBNext Multiplex Oligos for Illumina (New England BioLabs) by PCR. The quality and quantity of each amplified library were analyzed using the Bioanalyzer 2100 (Agilent Technologies), the High Sensitivity DNA kit (Agilent Technologies), and the NEBNext Library Quant Kit for Illumina (New England BioLabs). In total, 64 samples (16 samples per individual) were used for short-read sequencing.

## PacBio long-read sequencing and library preparation

To construct the reference genome of *S. laevis* and *S. leprosula*, high-molecular-weight DNA samples were extracted from branch 1–1 leaf materials of S1 and F1 individuals of each species, and sequenced using PacBio platforms. For *S. laevis*, library preparation and sequencing were performed at VBCF. The library was prepared using the SMRTbell express Kit (PacBio, Menlo Park, CA, USA), and sequenced on the Sequel platform with six SMRT cells (PacBio). For *S. leprosula*, library preparation and sequencing were performed by Macrogen Inc (Republic of Korea). The library for *S. leprosula* was prepared using the HiFi SMARTbell library preparation system (PacBio) according to the manufacturer's protocol, and was sequenced on the Sequel II platform (PacBio) with one SMRT cell.

## Genome assembly

The PacBio continuous long reads of *S. laevis* were assembled using Flye 2.7-b1587 (*Kolmogorov et al., 2019*) with 12 threads and with an estimated genome size of 350 Mbp. We subsequently used HyPo v1.0.3 (*Kundu et al., 2019*) for polishing the contigs. The Illumina read alignments provided to HyPo were created using Bowtie v2.3.4.3 (*Langmead and Salzberg, 2012*) with --very-sensitive option and using 32 threads. We used the Illumina reads from all branches of the individual S1 rather than utilizing exclusively those of branch 1–1, in order to capitalize on the increased aggregate sequencing depth.

The PacBio HiFi reads of *S. leprosula* with an average Quality Value (QV) 20 or higher were extracted, and subsequently assembled using Hifiasm 0.16.1-r375 (*Cheng et al., 2021*), with -z10 option and using 40 threads. The primary assembly of *S. leprosula* was used for further analysis. The quality and completeness of the genome assembly were assessed by searching for a set of 2326 core genes from eudicots_odb10 using BUSCO v5.3.0 (*Manni et al., 2021*) for each species (*Supplementary file 1d*).

## Genome annotation

We constructed repeat libraries of *S. laevis* and *S. leprosula* using EDTA v2.0.0 (*Ou et al., 2019*). Using the libraries, we ran RepeatMasker 4.1.2-p1 (*Smit et al., 2021*) with -s option and with Cross_match as a search engine, to perform soft-masking of trepetitive sequences in the genomes. The estimated percentages of the repetitive sequences were 42.4% for *S. laevis* and 39.5% for *S. leprosula* (*Supplementary file 1d*).

We ran BRAKER 2.1.6 (*Brůna et al., 2021*) to perform gene prediction by first incorporating RNA-seq data and subsequently utilizing a protein database, resulting in the generation of two sets of gene predictions for each species. To perform RNA-seq-based prediction, we mapped the RNA-seq reads (see RNA extraction in Methods section) to the genomes using HISAT 2.2.1 (*Kim et al., 2019*), with the alignments subsequently being employed as training data for BRAKER. For protein-based prediction, we used proteins from the Viridiplantae level of OrthoDB v10 (*Zdobnov et al., 2021*) as the training data.

The two sets of gene predictions were merged using TSEBRA (commit 0e6c9bf in the GitHub repository, *Hoff, 2022*; *Gabriel et al., 2021*) to select reliable gene predictions for each species. Although in principle TSEBRA groups overlapping transcripts and considers them as alternative spliced isoforms of the same gene, we identified instances where one transcript in a gene overlapped with another transcript in a separate gene. In such cases, we manually clustered these transcripts into the same gene.

We used EnTAP 0.10.8 (*Hart et al., 2020*) with default parameters for functional annotation. The databases employed were: UniProtKB release 2022_05 *Bateman, 2021*, NCBI RefSeq plant proteins release 215 (*O'Leary et al., 2016*), EnTAP Binary Database v0.10.8 (*Hart et al., 2020*) and EggNOG 4.1 (*Powell et al., 2014*). We constructed the standard gene model by utilizing the gene predictions of each species, eliminating any gene structures that lacked a complete ORF. Transcripts containing Ns were also excluded. Following the filtering process, the splice variant displaying the longest coding sequence (CDS) was selected as the primary isoform for each gene. The set of primary isoforms was used as the standard gene model.

## Genome size estimation

We estimated genome size of two species using GenomeScope (*Vurture et al., 2017*). We counted k-mer from forward sequence data of branch 1–1 from the S1 and F1 individuals using KMC 3 (*Kokot et al., 2017*) (k=21). The genome size and heterozygous ratio were estimated by best model fitting. Estimated genome sizes were 347 Mb for the slow-growing species and 376 Mb for the fast-growing species. These estimates were 8% and 7% smaller than the estimates obtained through flow cytometry (*Ng et al., 2016*), respectively. The genome size of the fast-growing species was nearly identical to that previously reported for *S. leprosula* in peninsular Malaysia (*Ng et al., 2021*).

## Genome synteny analysis

To investigate the syntenic relationship between *S. laevis* and *S. leprosula*, the synteny analysis performed using the MCScanX in TBtools-II (Toolbox for Biologists) v1.120 (https://github.com/CJ-Chen/TBtools/releases; *Chen, 2023*) with default parameters. For the synteny analysis, we selected 20 contigs from *S. leprosula* because these were the only ones that exhibited synteny blocks between the two species. 20 contigs covers more than 99.5% of the *S. leprosula* genome. The syntenic blocks spanning more than 30 genes were displayed in the synteny map (*Figure 1—figure supplement 3*).

## Somatic (intra-individual) SNV discovery

We filtered low quality reads out and trimmed adapters using fastp v22.0 (*Chen et al., 2018*) with following options: -q 20 n 10 t 1 T 1 l 20 w 16. The cleaned reads were mapped to the reference genome using bwa-mem2 22.1 (*Vasimuddin et al., 2019*) with default parameters. We removed PCR duplicates using fixmate and markdup function of samtools 1.13 (*Li et al., 2009*). The sequence reads were mapped to the reference genome, yielding average mapping rates of 91.61% and 89.5% for the slow- and fast-growing species, respectively. To identify reliable SNVs, we utilized two SNP callers bcftools mpileup (*Li et al., 2009*; *Li, 2011*) and GATK (4.2.4.0) HaplotypeCaller (*McKenna et al., 2010*) and extracted SNVs detected by both (*Figure 1—figure supplement 2*).

We first called SNVs with BCFtools 1.13 (*Danecek et al., 2021*) mpileup at three different thresholds; threshold 1 (T40): mapping quality (MQ)=40, base quality (BQ)=40; threshold 2 (T30): MQ = 30, BQ = 30; threshold 3 (T20): MQ = 20, BQ = 20. SNVs detected under each threshold were pooled for further analyses, with duplicates removed. We normalized indels using bcftools norm for vcf files. We removed indels and missing data using vcftools 0.1.16 (*Danecek et al., 2011*).

Second, we called SNVs using GATK (4.2.4.0) HaplotypeCaller and merged the individual gvcfs into a vcf file containing only variant sites. We removed indels from the vcf using the GATK SelectVariants. We filtered out unreliable SNVs using GATK VariantFiltration with the following filters: QD (Qual By Depth)<2.0, QUAL (Base Quality)<30.0, SOR (Strand Odds Ratio)>4.0, FS (Fisher Strand)>60.0, MQ (RMS Mapping Quality)<40.0, MQRankSum (Mapping Quality Rank Sum Test)<−12.5, ReadPosRankSum (Read Pos Rank Sum Test)<−8.0. After performing independent SNV calling for each biological replicate using each SNP caller, we extracted SNVs that were detected in both replicates for each SNP caller. We further extracted SNVs that were detected by both bcftools mpileup and GATK HaplotypeCaller (*Figure 1—figure supplement 2*) using Tassel5 (*Bradbury et al., 2007*) and a custom python script, generating potential SNVs for each threshold. Finally, SNVs detected at any of the three thresholds were extracted to obtain candidate SNVs. The number of SNVs at each filtering step can be found in *Supplementary file 1f*.

The candidate SNV calls were manually confirmed by two independent researchers using the IGV browser (*Robinson et al., 2017*). We removed sites from the list of candidates if there were fewer than five high-quality reads (MQ >20) in at least one branch sample among the 16 samples. After labeling branches carrying the called variant as somatic mutations, we compared the observed pattern with the genotyping call and extracted SNVs that were supported more than one read in both biological replicates (*Figure 3—figure supplement 2a*). We illustrated three types of false positive SNVs that were removed from the list of candidates in *Figure 3—figure supplement 2b–d*. The final set of SNVs can be found in *Supplementary file 1k*. Proportion of potential false positive and negative SNVs for each threshold are illustrated in *Figure 3—figure supplements 3 and 4*.

The NJ tree for each individual was generated using MEGA11 (*Tamura et al., 2021*) based on the matrix of the number of sites with somatic SNVs present between each pair of branches and edited using FigTree v1.4.4 (http://tree.bio.ed.ac.uk/software/figtree/). Most of the somatic SNVs were heterozygous, whereas 4% of the total SNVs (46/1136) were homozygous (*Supplementary file 1k*). The homozygous sites were treated as a single mutation due to the likelihood of a genotyping error being higher than the probability of two mutations occurring at the same site.

## Inter-individual SNV discovery

We also identified SNVs between pairs of individuals within each species as inter-individuals SNVs. The method for calling inter-individual SNVs was the same as for intra-individual SNVs, except that only threshold 2 (MQ = 30, BQ = 30) for BCFtools 1.13 (*Danecek et al., 2021*) was used. We extracted SNVs that are present in all branches within an individual using Tassel5 (*Bradbury et al., 2007*). To exclude ambiguous SNV calls, we removed SNVs within 151 bp of indels that were called with BCFtools 1.13 (*Danecek et al., 2021*) with the option of threshold 2. We eliminated SNVs within 151 bp of sites with a depth value of zero that occur in more than ten consecutive sites. We also removed SNVs that had a depth smaller than five or larger than $d + 3\sqrt{d}$, where $d$ represents the mean depth of all sites (*Li, 2014*). Due to the large number of candidates for inter-individual SNVs, the manual checking process was skipped.

## Somatic SNVs confirmation by amplicon sequencing

We verified the reliability of the final set of somatic SNVs by amplicon sequencing approximately 5% of the SNVs in *S. laevis* (31 and 10 SNVs for S1 and S2, respectively). We used multiplexed phylogenetic marker sequencing method MPM-seq (*Suyama et al., 2022*) with modifications to the protocol as follows: to amplify 152–280 bp fragments, the first PCR primers comprising tail sequences for the second PCR primers were designed on the flanking regions of each SNV. The first PCR was conducted using the Fast PCR cycling kit (Qiagen, Düsseldorf, Germany) under the following conditions: an initial activation step at 95 °C for 5 min, followed by 30 cycles of denaturation at 96 °C for 5 s, annealing at 50/54/56 °C for 5 s, and extension at 68 °C for 10 s. This was followed by a final incubation at 72 °C for

1 min. Subsequent next-generation sequencing was performed on an Illumina MiSeq platform using the MiSeq Reagent Kit v2 (300 cycles: Illumina).

Amplicon sequencing reads were mapped to the reference genome using bwa-mem2 22.1 (*Vasimuddin et al., 2019*) with default parameters. Using bcftools mpileup (*Danecek et al., 2021*), we called the genotypes of all sites on target regions and eliminated candidate sequences with MQ and BQ less than 10. The final set of sites selected for confirmation consisted of 24 for the S1 individual and 9 for the S2 individual. We manually confirmed the polymorphic patterns at the target sites using the IGV browser (*Robinson et al., 2017*). If the alternative allele was present or absent in all eight branches in the amplicon sequence, the site was determined as fixed. The site was determined as mismatch if the difference of polymorphic patterns between the somatic SNV calls and amplicon sequence was supported by more than four reads per branch. The sites that were neither fixed nor mismatched were determined as true. 94% (31/33) of SNVs at the final target sites, with 22 out of 24 mutations on S1 and all 9 mutations on S2, were confirmed to exhibit a polymorphic pattern that exactly matched between the somatic SNV calls and amplicon sequence (*Supplementary file 1g*). It is important to note that the SNVs that were not matched with amplicon sequencing data could potentially represent true somatic mutations. This discrepancy could be attributed to a low allele frequency, where the call is not identified as heterozygous despite the presence of a true mutation.

## Somatic mutation rates per growth and per year

To estimate the somatic mutation rate per nucleotide per growth ($\mu_g$), a linear regression analysis of the number of somatic SNVs against the physical distance between sampling positions within an individual was conducted using the lm package, with an intercept of zero, in R version 3.6.2. The somatic mutation rate per nucleotide per growth was estimated as:

$$\mu_g = \frac{b}{2 \times R},$$

where $b$ indicates the slope of linear regression and $R$ denotes the number of callable sites, respectively. Note that the denominator includes a factor of two due to diploidy. A site was considered callable when it passed the filters as the polymorphic sites, that is, a mapping quality of at least 40 using GATK, a mapping quality of at least 20 using BCFtools, and a depth greater than or equal to 5. This resulted in 388,801,756 and 320,739,335 base pairs for S1 and S2 and 327,435,618 and 263,488,812 base pairs for F1 and F2, respectively.

The somatic mutation rate per nucleotide per year ($\mu_y$) was estimated as:

$$\mu_y = \frac{M}{2 \times R \times A}$$

Here, $M$ indicate the total number of SNVs accumulated from the base (ID 0 in *Figure 1a*; *Supplementary file 1b*) to the branch tip and $A$ represents tree age, respectively. $R$ denotes the number of callable sites that was also used to estimate $\mu_g$. Because there are seven branch tips for each tree (*Figure 1a*), we estimated $\mu_y$ for each of branch tips and then calculated the mean and 95% confidence interval for each tree (*Supplementary file 1h*).

## Mutational spectrum

Mutational spectra were derived directly from the reference genome and alternative alleles at each variant site. There are a total of six possible classes of base substitutions at each variant site: A:T>G:C (T>C), G:C>A:T (C>T), A:T>T:A (T>A), G:C>T:A (C>A), A:T>C:G (T>G), and G:C>C:G (C>G), By considering the bases immediately 5′ and 3′ to each mutated base, there are a total of 96 possible mutation classes, referred to as triplets, in this classification. We used seqkit (*Shen et al., 2016*) to extract the triplets for each variant site. To count the number of each triplet, we used the Wordcount tool in the EMBOSS web service (https://www.bioinformatics.nl/cgi-bin/emboss/wordcount). We calculated the fraction of each mutated triplet by dividing the number of mutated triplets by the total number of triplets in the reference genome.

We compared the mutational signatures of our tropical trees to those of single-base substitution (SBS) signatures in human cancers using Catalogue Of Somatic Mutations In Cancer (COSMIC) compendium of mutation signatures (COSMICv.2 *Alexandrov et al., 2013*; *Nik-Zainal et al., 2016*;

*Alexandrov et al., 2020*; *Greenman et al., 2006*; *Martincorena et al., 2017*, available at https://cancer.sanger.ac.uk/cosmic/signatures_v2). Cosine similarity was calculated between each tropical tree species and each SBS signature in human cancers.

## Testing selection of somatic and inter-individual SNVs

To test whether somatic and inter-individual SNVs are subject to selection, we calculated the expected rate of non-synonymous mutation. For the CDS of length $L_{cds}$, there are possible numbers of mutations of length of $3L_{cds}$ (*Figure 4—figure supplement 1*). We classified all possible mutations into three types based on the codon table: synonymous, missense, and nonsense (*Figure 4—figure supplement 1*). Each type of mutation was counted for each of the six base substitution classes (*Figure 4—figure supplement 1*). We generated count tables based on two distinct categories of CDS: those that included all isoforms and those that only encompassed primary isoforms (*Supplementary file 1l*). As the two tables were largely congruent, we employed the version which included all isoforms of CDS.

Using the count table and background mutation rate for each category of substitution class, we calculated the expected number of synonymous ($\lambda_S$) and non-synonymous mutations ($\lambda_N$) (*Figure 4—figure supplement 1*). As a background mutation rate, we adopted the observed somatic mutation rates in the six substitution classes in the intergenic region (*Supplementary file 1m*), assuming that the intergenic region is nearly neutral to selection. Because the number of nonsense somatic mutation is small, we combined missense and nonsense mutations as non-synonymous. The intergenic regions were identified as the regions situated between 1 kbp upstream of the start codon and 500 bp downstream of the stop codon. Expected rate of synonymous mutation ($p_N$) is given as $\lambda_N/(\lambda_S+\lambda_N)$. Given the observed number of non-synonymous and synonymous mutations, we rejected the null hypothesis of neutral selection using a binomial test with the significance level of 5% (*Supplementary file 1j*). We used the package binom.test in R v3.6.2.

We also used the observed somatic mutation rate in the whole genome (*Supplementary file 1m*), including genic and intergenic regions, as the background mutation rate and confirmed the robustness of our conclusion (*Supplementary file 1j*). The somatic mutation rates in the intergenic region and the whole genome were calculated for each species by pooling the data from two individuals (*Supplementary file 1m*). While cancer genomics studies have accounted for more detailed context-dependent mutations, such as the high rate of C>T at CpG dinucleotides (*Greenman et al., 2006*) or comprehensive analysis of 96 possible substitution classes in triplet context (*Martincorena et al., 2017*), the number of SNVs in our tropical trees is too small to perform such a comprehensive analysis. Therefore, we used the relatively simple six base substitution classes. The genes with somatic SNVs can be found in *Supplementary file 1n*.

## Acknowledgements

The authors thank to M Ohno for her insightful discussion, M Seki for his assistance with statistical analysis, S K Hirota for his technical support in molecular experiments, and Y Ikezaki for her support in synteny analyses. We also thank Y Iwasa, H Tachida, M M Manuel, N Spisak, M Przeworski and M, Nordborg for their very insightful comments on the initial draft of our manuscript.

## Additional information

### Funding

| Funder | Grant reference number | Author |
|---|---|---|
| Japan Society for the Promotion of Science | JP17H06478 | Akiko Satake |
| Japan Society for the Promotion of Science | JP22H04925 | Masahiro Kasahara |

The funders had no role in study design, data collection and interpretation, or the decision to submit the work for publication.

**Author contributions**
Akiko Satake, Conceptualization, Resources, Formal analysis, Supervision, Funding acquisition, Validation, Methodology, Writing – original draft, Project administration, Writing – review and editing; Ryosuke Imai, Data curation, Software, Formal analysis, Validation, Investigation, Visualization, Writing – review and editing; Takeshi Fujino, Data curation, Software, Writing – review and editing; Sou Tomimoto, Formal analysis, Validation, Visualization; Kayoko Ohta, Almudena Molla Morales, Viktoria Nizhynska, Investigation; Mohammad Na'iem, Sapto Indrioko, Widiyatno Widiyatno, Susilo Purnomo, Naoki Tani, Resources; Yoshihisa Suyama, Resources, Investigation, Writing – review and editing; Eriko Sasaki, Formal analysis, Validation, Visualization, Writing – review and editing; Masahiro Kasahara, Data curation, Software, Funding acquisition, Writing – review and editing

**Author ORCIDs**
Akiko Satake ⓘ https://orcid.org/0000-0002-0831-8617
Sou Tomimoto ⓘ https://orcid.org/0009-0005-8376-7104
Eriko Sasaki ⓘ https://orcid.org/0000-0002-0878-364X

Reviewer #1 (Public review): https://doi.org/10.7554/eLife.88456.3.sa1
Reviewer #2 (Public review): https://doi.org/10.7554/eLife.88456.3.sa2
Reviewer #3 (Public review): https://doi.org/10.7554/eLife.88456.3.sa3
Author response https://doi.org/10.7554/eLife.88456.3.sa4

## Additional files

**Supplementary files**
• Supplementary file 1. Supplementary tables. (**a**) Mean annual increment (MAI) of diameter at breast height (DBH). (**b**) Matrix of physical distances (m) between sampling positions and the number of SNVs indicated in parentheses. (**c**) Summary statistics of the studied trees. Height and DBH were directly measured for two individuals of *S. laevis* and *S. leprosula*. Age was estimated as DBH divided by a mean annual increment (MAI). (**d**) Summary statistics of genome assemblies for *S. laevis* and *S. leprosula*. We assembled the genome using DNA extracted from the apical leaf at branch 1–1 of the tallest individual of each species (S1 and F1). Summary statistics of genome assemblies are listed here. (**e**) Summary statistics of whole genome sequencing. (**f**) The number of candidate SNVs during each step of the filtering process. (**g**) Assessment of candidate SNVs using amplicon sequencing. (**h**) Somatic mutation rates. The somatic mutation rate per nucleotide per meter was estimated as $\mu_g = \frac{b}{2 \times R}$, where $b$ indicates the slope of linear regression. The somatic mutation per nucleotide per year ($\mu_y$) was estimated as $\mu_y = \frac{M}{2 \times R \times A}$, where $M$ indicates the total number of SNVs accumulated from the base to the branch tip and $A$ represents tree age, respectively. $R$ denotes the number of callable sites. (**i**) Cosine similarity of mutation spectra between *Shorea* trees and humans. (**j**) Results of the binomial test for selection on somatic and inter-individual SNVs. To test whether somatic and inter-individual SNVs are subject to selection, we calculated the expected rate of non-synonymous mutation. Given the observed number of non-synonymous and synonymous mutations, we rejected the null hypothesis of neutral selection using a binomial test with the significance level of 5%. $p_N$_expected and $p_N$_observed represent the expected and observed rate of non-synonymous substitutions. (**k**) The final set of SNVs. (**l**) Fractions of synonymous, missense, and nonsense substitutions. (**m**) Somatic mutation rates for six substitution classes. Somatic mutation rates for six substitution classes were calculated based on the observed number of SNVs both from the intergenic region and the whole genome. S1 +S2 and F1 +F2 represent the use of pooled data from two individuals for each species: *S. laevis* (S1, S2) and *S. leprosula* (F1, F2). The values based on the pooled data (indicated in bold type) were used to calculate the expected rate of non-synonymous mutation. (**n**) List of genes with somatic SNVs.

• MDAR checklist

**Data availability**
The raw sequencing data, the genome assembly, and the gene annotation are available at DDBJ under accessions PRJDB14538 for *S. laevis* and PRJDB15012 for *S. leprosula*.The codes for the bioinformatics pipeline to process whole genome sequencing data is available from https://github.com/ku-biomath/Shorea_mutation_detection (copy archived at *ku-biomath, 2023*).

The following datasets were generated:

| Author(s) | Year | Dataset title | Dataset URL | Database and Identifier |
|---|---|---|---|---|
| Satake A, Imai R, Fujino T, Tomimoto S, Ohta K, Na'iem M, Indrioko S, Widiyatno PS, Mollá-Morales A, Nizhynska V, Tani N, Suyama Y, Sasaki E, Kasahara M | 2023 | Genetic mosaicism and somatic mutation rate in tropical trees | https://ddbj.nig. ac.jp/search/ entry/bioproject/ PRJDB14538 | DNA Data Bank of Japan, PRJDB14538 |
| Satake A, Imai R, Fujino T, Tomimoto S, Ohta K, Na'iem M, Indrioko S, Widiyatno PS, Mollá-Morales A, Nizhynska V, Tani N, Suyama Y, Sasaki E, Kasahara M | 2023 | Genetic mosaicism and somatic mutation rate in tropical trees | https://ddbj.nig. ac.jp/search/ entry/bioproject/ PRJDB15012 | DNA Data Bank of Japan, PRJDB15012 |

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
