## [Editor Report · eLife assessment]

Satake and colleagues' **important** study elucidates somatic mutation processes in plants, demonstrating that in two tropical trees, mutation rates correlate with age, not growth rates. Their **convincing** evidence shows that many mutations do not align with cell divisions, suggesting many somatic mutations are generated in a replication-independent manner. This study represents a significant step towards advancing our understanding of plant development and the patterns and inheritance of mutations. This significant research is poised to engage a diverse array of scholars in plant evolution and development.

---

## [Referee Report · Reviewer #1 (Public review)]

In this study, Satake and colleagues endeavored to explore the rates and patterns of somatic mutations in wild plants, with a focus on their relationship to longevity. The researchers examined slow- and fast-growing tropical tree species, demonstrating that slow-growing species exhibited five times more mutations than their fast-growing counterparts. The number of somatic mutations was found to increase linearly with branch length. Interestingly, the somatic mutation rate per meter was higher in slow-growing species, but the rate per year remained consistent across both species. A closer inspection revealed a prevalence of clock-like spontaneous mutations, specifically cytosine-to-thymine substitutions at CpG sites. The author suggested that somatic mutations were identified as neutral within an individual, but subject to purifying selection when transmitted to subsequent generations. The authors developed a model to assess the influence of cell division on mutational processes, suggesting that cell-division independent mutagenesis is the primary mechanism.

The authors have gathered valuable data on somatic mutations, particularly regarding differences in growth rates among trees. Their meticulous computational analysis led to fascinating conclusions, primarily that most somatic mutations accumulate in a cell-division independent manner. The discovery of a molecular clock in somatic mutations significantly advances our comprehension of mutational processes that may generate genetic diversity in tropical ecosystems. The interpretation of the data appears to be based on the assumption that somatic mutations can be effectively transmitted to the next generation unless negative selection intervenes. However, accumulating evidence suggests that plants may also possess "effective germlines," which could render the somatic mutations detected in this study non-transmittable to progeny. Incorporating additional analyses/discussion in the context of plant developmental biology, particularly recent studies on cell lineage, could further enhance this study.

Specifically, several recent studies address the topics of effective germline in plants. For instance, Robert Lanfear published an article in PLoS Biology exploring the fundamental question, "Do plants have a segregated germline?" A study in PNAS posited that "germline replications and somatic mutation accumulation are independent of vegetative life span in Arabidopsis." A phylogenetic-based analysis titled "Rates of Molecular Evolution Are Linked to Life History in Flowering Plants" discovered that "rates of molecular evolution are consistently low in trees and shrubs, with relatively long generation times, as compared with related herbaceous plants, which generally have shorter generation times." Another compelling study, "The architecture of intra-organism mutation rate variation in plants," published in PLoS Biology, detected somatic mutations in peach trees and strawberries. Although some of these studies are cited in the current work, a deeper examination of the findings in relation to the existing literature would strengthen the interpretation of the data.

---

## [Referee Report · Reviewer #2 (Public review)]

In this manuscript, the authors used an original empirical design to test if somatic mutation rates are different depending on the plant growth rates. They detected somatic mutations along the growth axes of four trees - two individuals per species for two dipterocarp tree species growing at different rates. They found here that plant somatic mutations are accumulated are a relatively constant rate per year in the two species, suggesting that somatic mutation rates correlate with time rather than with growth, i.e. the number of cell divisions. The authors then suggest that this result is consistent with a low relative contribution of DNA replication errors (referred to as α in the manuscript) to the somatic mutation rates as compared to the other sources of mutations (β). Given that plants - in particular, trees - are generally assumed to deviate from the August Weismann's theory (a part of the somatic variation is expected to be transmitted to the next generation), this work could be of interest for a large readership interested by mutation rates as a whole, since it has implications also for heritable mutation rates too. In addition, even if this is not discussed, the putatively low contribution of DNA replication errors could help to understand the apparent paradox associated to trees. Indeed, trees exhibit clear signatures of lower molecular evolution (Lanfear et al. 2013), therefore suggesting lower mutation rates per unit of time. Trees could partly keep somatic mutations under control thanks to a long-term evolution towards low α values, resulting in low α/β ratios as compared to short-lived species. I therefore consider that the paper tackles a fundamental albeit complex question in the field.

Overall, I consider that the authors should clearly indicate the weaknesses of the studies. For instance, because of the bioinformatic tools used, they have reasonably detected a small part of the somatic mutations, those that have reached a high allele frequency in tissues. Mutation counts are known to be highly dependent on the experimental design and the methods used. Consequently, (i) this should be explicit and (ii) a particular effort should be made to demonstrate that the observed differences in mutation counts are robust to the potential experimental biases. This is important since, empirically, we know how mutation counts can vary depending on the experimental designs. For instance, a difference of an order of magnitude has been observed between the two papers focusing on oaks (Schmid-Siegert et al. 2017 and Plomion et al. 2018) and this difference is now known to be due to the differences in the experimental designs, in particular the sequencing effort (Schmitt et al. 2022).

Having said that, my overall opinion is that (i) the authors have worked on an interesting design and generated unique data, (ii) the results are probably robust to some biases and therefore strong enough (but see my comments regarding possible improvements), (iii) the interpretations are reasonable and (iv) the discussion regarding the source of somatic mutations is valuable (even if I also made some suggestions here also).

---

## [Referee Report · Reviewer #3 (Public review)]

In animals, several recent studies have revealed a substantial role for non-replicative mutagenic processes such as DNA damage and repair rather than replicative error as was previously believed. Much less is known about how mutation operates in plants, with only a handful of studies devoted to the topic. Authors Satake et al. aimed to address this gap in our understanding by comparing the rates and patterns of somatic mutation in a pair of tropical tree species, slow-growing Shorea lavis and fast-growing S. leprosula. They find that the yearly somatic mutation rates in the two species is highly similar despite their difference in growth rates. The authors further find that the mutation spectrum is enriched for signatures of spontaneous mutation and that a model of mutation arising from different sources is consistent with a large input of mutation from sources uncorrelated with cell division. The authors conclude that somatic mutation rates in these plants appears to be dictated by time, not cell division numbers, a finding that is in line with other eukaryotes studied so far.

In general, this work shows careful consideration and study design, and the multiple lines of evidence presented provide good support for the authors' conclusions. In particular, they use a sound approach to identify rare somatic mutations in the sampled trees including biological replicates, multiple SNP-callers and thresholds, and without presumption of a branching pattern.

Inter-species comparisons of absolute mutation rates is challenging. This is largely due to differences in SNP-calling methods and reference genome quality leading to variable sensitivity and specificity in identifying mutations. By applying their pipeline consistently across both species, the authors provide confidence in the comparative mutation rate results. Moreover, the presented false negative and false positive rate estimates for each species would apparently have minimal impact on the overall findings.

Despite the overall elegance of the authors' experimental setup, one methodological wrinkle warrants consideration. The authors compare the mutation rate per meter of growth, demonstrating that the rate is higher in slow-growing S. laevis: a key piece of evidence in favor of the authors' conclusion that somatic mutations track absolute time rather than cell division. To estimate the mutation rate per unit distance, they regress the per base-pair rate of mutations found between all pairwise branch tips against the physical distance separating the tips (Fig. 2a). While a regression approach is appropriate, the narrowness of the confidence interval is overstated as the points are not statistically independent: internal branches are represented multiple times. (For example, all pairwise comparisons involving a cambium sample will include the mutations arising along the lower trunk.) Regressing rates and lengths of distinct branches might be more appropriate. Judging from the data presented, however, the point estimates seem unlikely to change much.

This work deepens our understanding of how mutation operates at the cellular level by adding plants to the list of eukaryotes in which many mutations appear to derive from non-replicative sources. Given these results, it is intriguing to consider whether there is a fundamental mechanism linking mutation across distantly related species. Plants, generally, present a unique opportunity in the study of mutation as the germline is not sequestered, as it is in animals, and thus the forces of both mutation and selection acting throughout an individual plant's life could in principle affect the mutations transmitted to seed. The authors touch on this aspect, finding no evidence for a reduction in non-synonymous somatic mutations relative to the background rate, but more work-both experimental and observational-is needed to understand the dynamics of mutation and cell-competition within an individual plant. Overall, these results open the door to several intriguing questions in plant mutation. For example, is somatic mutation age-dependent in other species, and do other tropical plants harbor a high mutation rate relative to temperate genera? Any future inquiries on this topic would benefit from modeling their approach for identifying somatic mutations on the methods laid out here.

---

## [Author Response]

The following is the authors’ response to the original reviews.

**Reviewer #1**
1. Here are a few sentences that could potentially benefit from further discussion, particularly in the context of the plant developmental framework of an effective germline. It is important to note that the idea of an effective germline is supported by many, but not all, scientists. Nevertheless, as long as this concept remains relevant, a discussion based on it may be appropriate.

The early establishment of germlines during development is crucial in addressing the impact of somatic mutation on the next generation. To emphasize this aspect, we have included an additional sentence addressing this point in ll. 242–244.

1. Lines 161-163: The suggestion that long-lived tropical trees do not necessarily suppress somatic mutation rates to the same extent as their temperate counterparts might warrant additional examination.

We have revised our statement to present a more balanced perspective, and we have also included a sentence to emphasize the importance of conducting further studies in future.

1. Lines 200-202: The observation of potential influences of GC-biased gene conversion during meiosis or biased purifying selection for C>T inter-individual nucleotide substitutions could be further elaborated upon.

Our data does not provide enough information to delve into a more detailed discussion regarding GC-biased gene conversion during meiosis or biased purifying selection for C>T substitution. However, future studies that obtain genome sequences from somatic cells, male or female gametophytes, and offspring (such as seeds or seedlings) would offer opportunities to assess these phenomena.

1. Line 245: The statement "somatic mutations can be transmitted to seeds" might be correct, but it would be helpful to explore the extent to which this occurs.

In response to the comment from Reviewer 1 (#4) and 2 (#16), we have decided to remove the discussion about the heritability of somatic mutations in next generation. We have completely rewritten the final paragraph to discuss the possibility of a disparity in the relationship between lifespan and somatic mutation rates between plants and animals.

**Reviewer #2**
1. l. 108- 115: The authors seem to have made a really great work at assembling and annotating two reference genomes. Even if this does not represent the main result of the manuscript, these genomic resources are a plus for the community, especially given that reference genomes from tropical trees are known to be underrepresented in the literature (e.g. Plomion et al. 2016). The authors have made the particular effort of generating two high-quality reference genome assemblies for two species of the same genus, including one with an excellent contiguity. Even if they do not explicitly indicate the divergence time between the two species, it is clear that the cheapest solution would have been to map the reads of the two species against a single assembly, but this could have generated some biases. So by generating two de novo assemblies, the authors have used here the best design possible to control for some potential biases for the detection of somatic mutations. However, given the interests these two assemblies represent by themselves, I consider that a couple of additional investigations could have been made on local synteny and orthologous genes in particular. Thanks to whole-genome alignments and orthology (e.g. Lovell et al. 2022), they could have generated more general information regarding the two assembles and investigated additional questions regarding mutations, e.g. mutations in collinear / non-collinear (if any) segments, intensity of purifying selection (or neutral evolution) at single vs. multiple copies or between shared vs. private genes, etc.

To address the comment by Reviewer 2, we performed synteny analysis using the MCScanX in TBtools-II and added Supplementary Figure 3 to illustrate conserved synteny relationship between S. laevis and S. leprosula. Detecting selection in the genome will be a future study as our current data are not sufficient for the aim because of limited number of individuals (n = 2 for each species).

1. l. 123-124. Here, the authors indicate that they have "validated" 93.9% of the mutations. It would be more accurate to indicate that they have "validated" 31/33 mutations (94%), 22/24 mutations on S1 and 9/9 on S2 (Table S5). Can the authors indicate why no somatic mutations from the F1 and F2 were tested? According to me, the use of the word "validation" is not totally accurate (see also Schmitt et al. 2022), since amplicon sequencing can be viewed as a kind of validation but it doesn't represent a complete validation since it represents new sequencing data that are mapped against the same reference assembly, in such a way that we could always imagine that the same biases are at play, leading to a similarly false positive call. Reciprocally, a "non-validated" mutation could be associated to a mutation that is at a too low allele frequency, at least after amplification, in such a way that the call is not heterozygous despite the fact that the mutation is real. I think that another terminology than "validated" could be used, plus one or two sentences explaining this degree of complexity.

To improve the clarity of the statement, we have modified the sentence as follows: We conducted an independent evaluation of a subset of the inferred single nucleotide variants (SNVs) using amplicon sequencing. Our analysis demonstrated accurate annotation for 31 out of 33 mutations (94% overall), with 22 out of 24 mutations on S1 and all 9 mutations on S2 (Supplementary Table 5).”

While we did not conduct additional assessments using F1 and F2, we anticipate a similar high level of agreement between the somatic SNV calls and amplicon sequencing in these trees. We have included sentences in the Materials and Methods section to elucidate the challenges involved in validating true somatic mutations.

1. l. 135-137 the reasoning appears to be quite circular to me. As indicated by the authors in the line just before, an incongruent pattern could also be explained biologically, in such a way that the overall congruency between the phylogenetic tree and the tree architecture cannot be considered as a way to prove the reliability of the detection. In some species, it seems clear that the phylogenetic tree do not seem to follow the plant architecture (Zahradnikova et al. 2020) in such a way that we should argue to not consider the plant architecture in the design and not consider this represents either a way to validate mutations or a way to validate the methodological framework. I suggest removing this sentence.

We have removed the sentence as suggested by Reviewer 2.

1. l. 150. It seems that the differences in length and diameter between the two species come from two different studies and therefore that no statistical test has been performed to test its significance.

We agree with Reviewer 2. To clarify this point, we have replaced “significantly” with “substantially” in the revised text.

1. l. 156-159: the same sentence is repeated twice.

We have removed the repeated sentence.

1. l. 159-161: Comparing somatic mutation rates between studies is difficult. It is too sensitive to the methodology used, here again see Schmitt et al. 2022. I propose to remove these two sentences. It represents an interesting working hypothesis but would require a better design, or at least, to reanalyze all the data with the same pipeline.

We have toned down our statement, and added a sentence that additional studies are required to compare somatic mutation rates among trees in tropical, temperate, and boreal regions, employing standardized methodologies.

1. l. 171-175: Here I am wondering if the authors could provide more information regarding the enrichment at CpG sites? I suggest first estimating the proportion of CpG sites thanks to the two genome assemblies and then using this information as a way to weight the results and therefore to estimate the level of enrichment of mutations at CpG sites.

In response to the comment by Reviewer 2, we first determined the proportion of CpG sites as 0.030 and 0.028 for S. laevis and S. leprosula, respectively, based on the triplet matrix using the reference genome of each species. Subsequently, we estimated the proportion of somatic mutations at CpG sites. The results revealed a 4.54-fold and 3.53-fold increase in somatic mutations at CpG sites for S1 and S2, and a 3.38-fold and 2.56-fold increase for F1 and F2, respectively. We have incorporated this finding into ll. 172–175.

1. l. 176-187. Interesting comparison and insights. You could also indicate that SBS5 is also detected in all human cancers too. So the detection of SBS1 and SBS5 signatures indeed suggest some shared mutation biases. Note that in humans, a specific signature of UV is associated to TCG -> TTG mutations (Martincorena & Campbell, 2015). It seems that there is a substantial difference in the mutation spectra between the two trees for this specific category, note sure if this difference could be associated to UV.

We slightly modified the sentence to indicate that SBS5 is also detected in all human cancers. We are very interested in the potential impact of UV on somatic mutations in tropical trees, considering the high levels of UVR in the tropics. Conducting a comparative analysis of the mutational spectrum among trees inhabiting diverse UVR environments would provide valuable insights to substantiate this hypothesis.

1. l. 206: I rather suggest "the somatic mutation rate per year is roughly the same, suggesting that somatic mutations rates are independent of growth rate".

In response to the suggestion from Reviewer 2, we have revised the sentence as follows: "The somatic mutation rate per year remains largely consistent, indicating that somatic mutation rates are independent of the growth rate."

1. l. 207-232: Here, It is the section looks a mixture between a result and a discussion. I guess the authors consider here that it remains a verbal model at this stage and it therefore represents more a discussion. If so, I agree but it could be good to discuss more this part, in particular to know how this model could be improved and empirically tested.

The argument based on the model will be more accurate when the cell cycle duration can be directly estimated for each tree. We have added this explanation in the revised text.

1. l. 238-239: The parallel drawn with the molecular clock is interesting but according to me, it remains a working hypothesis at this stage, since it is not validated outside the two focal species. I encourage the readers to continue to work on this question and to investigate also some annual plants for instance in the future (assuming that they have a higher α) in order to be able to derive a global model. In addition, even if I consider that the authors use and interpret this parallel wisely, I consider that the use of this terminology could be misleading for some readers. That's why I also suggest removing "molecular clock" from the title and using a more explicit one, e.g. "Somatic mutation rates scale with time not growth rate in dipterocarp trees".

We agree with Reviewer 2. We have changed the title to “Somatic mutation rates scale with time not growth rate in long-lived tropical trees.”

1. l. 245-249: The results rather suggest that (i) there is little diversity due to somatic mutations and that (ii) most heritable non-synonymous mutations are deleterious and therefore purged from the population. So rather than this last section of this discussion that has little interest and could be quite debatable, I consider that the authors could extend their discussion, e.g. the differences with somatic mutations in mammals (recently, Cagan and coauthors (2022) demonstrated that somatic mutation rates are inversely correlated with lifespan in mammals) or the overall low rate of molecular evolution in trees could be some directions. But there are many others.

We have completely rewritten the final paragraph to propose the possibility of a disparity in the relationship between lifespan and somatic mutation rates between plants and animals, rather than discussing the heritability of somatic mutation in next generation.

1. l. 570-571: I guess, the reader should understand here "fixed at the heterozygous state"

To avoid confusion, we have modified the text as follows: “If the alternative allele was present or absent in all eight branches in the amplicon sequence, the site was determined as fixed within an individual tree.” We have also removed “heterozygote” in Supplementary Figure 5.

1. Fig. 4d. the y-axis would be easier to interpret by writing "Delta Inter-individual vs. Somatic SNPs" and/or by adding arrows on the right margin of the plot to indicate the directions with some short sentences such as "more somatic mutations observed than expected assuming the inter-individual comparison", "less somatic mutation than expected". According to me, some statistical tests are lacking here. Are the differences in the mutation spectra significant given the relatively limited amount of somatic mutations detected?

We have added short sentences explaining the directions.

1. Supplementary Tables (excel file): please correct the typos. There are many on these supplementary tables.

We carefully checked supplementary tables and corrected the typos.

**Reviewer #3**
1. To estimate false negative rates, the authors might consider using mutation insertion tools such as Bamsurgeon (https://github.com/adamewing/bamsurgeon) to create simulated mutations. Alternatively, one could assess the calling rate of high-confidence SNPs that differ between individuals of the same species to get at the FNR.

We agree with Reviewer 3. To calibrate our pipeline, we previously performed simulation to estimate the false negative and positive rates in different tree species (Betula platyphylla) using wgsim v0.1.11 (https://github.com/lh3/wgsim). Based on our simulations, we found that the false negative and false positive rates were very low, averaging at 0.050 and 0.046, respectively. It is important to note that the estimated false positive rate obtained from the simulation data was substantially lower than the proportion of potential false positive SNVs (as shown in Supplementary Fig. 5). This observation suggests that simulation-based evaluation of the false positive rate is not reliable, at least for the tree species we studied. Similarly, the same argument could be applied to the false negative rate. Therefore, we conclude that the simulation-based analysis for estimating false positive and false negative rates is not informative for our study.

The rate of true-positive or false-negative mutation calls can be estimated only when the true mutational status is known, but the data are not currently available. However, under the assumption that the final set of SNVs represents true somatic mutations, we were able to calculate the potential false negative rate. Our findings indicate that this rate is low, specifically less than 10%, when using less stringent filtering thresholds such as BQ20 and MQ20. While these estimated values may not precisely represent the true false negative rate, we included them as potential false negative rates in Supplementary Figure 7 of the revised manuscript. This information provides additional insights into the performance of our pipeline under different filtering thresholds and contributes to the overall assessment of our study.

1. It may be interesting to examine the mutation trees for constancy (or not) in mutation rate per meter. Examining Figure 1, it appears that the number of mutations near the crown "4" node is consistently higher than in nearby nodes (3-1 and 3-2).

We calculated the branch-level increment of SNVs per meter by dividing the number of single nucleotide variations (SNVs) by the physical distance. Our analysis revealed a slight increase in the number of SNVs per meter as the branch position became higher in S. laevis, as shown in Author response table 1. However, this trend was not clearly observed in S. leprosula. We found this observation in S. laevis intriguing, particularly because our recent analysis (Tomimoto et al., in preparation) demonstrated that genetic distance increases in branch pairs located in the upper part of a tree. This was elucidated through a mathematical model that describes the dynamics of the stem cell population during elongation and branching. We opted not to delve further into the findings in the current manuscript, as this topic will be extensively investigated in a future study.

**Author response table 1. sa4table1:** The branch-level increment of SNVs per meter.

	Branch ID	0	1_1	12		2.1		22		3_1		32	
	1-1	3.23											
S. laevis	1.2	5.18	13.32										
	2.1	4.01	4.67		6.42								
S1	2.2	3.78	4.42		6.19		7.18						
	3.1	4.33	5.06		6.89		6.81		6.43				
	3.2	4.30	5.02		6.85		6.75		6.38		10.82		
	4	4.14	4.77		6.43		5.95		5.86		7.43		7.41
	1.1	2.37											
	1.2	2.36	1.96										
	2-1	3.11	2.86		2.84								
latevis	2.2	3.05	2.79		2.77		3.91						
S2	3.1	2.61	2.23		1.86		3.45		3.36				
	3.2	2.59	2.20		1.84		3.40		3.31		3.92		
	4	2.77	2.43		2.41		3.73		3.63		3.74		3.67
	1.1	1.21											
	1.2	1.11	1.18										
- leprosula	2-1	1.50	1.35		1.19								
F1	_(2)^(2)2	1.57	1.46		1.28		2.22						
	3=1	1.27	1.05		0.96		1.41		1.52				
	32	1.19	0.95		0.87		1.27		1.38		2.41		
	4	1.07	0.83		0.78		1.05		1.14		0.97		0.82
	1.1	0.83											
	12	0.55	2.07										
	2-1	0.34	1.10		0.54								
S. leprosula	2.2	0.39	1.19		0.62		1.25						
F?	3+1	0.44	1.24		0.68		0.46		0.53				
	3.2	0.47	1.32		0.74		0.50		0.58		1.15		
	4	0.60	1.57		0.95		0.71		0.78		0.97		1.05

1. Line 150: Use of "significantly different" is confusing as the phrase is usually reserved for statistical significance. Consider replacing with "substantially different."

We have replaced “significantly” with “substantially” in the revised text.

1. In the Discussion, a clearer explanation of the assumptions that underlie the authors' reasoning would be welcome: e.g., constancy in mutation rate per meter within an individual tree. In particular, the authors assume that mutations that are seen in one leaf and not in another cannot have predated the most recent common meristematic node linking the two leaves. Is this a reasonable assumption? Since the meristem is multicellular, is it possible for a mutation to have arisen earlier in development and "assorted" into one cell lineage but not another?

We greatly appreciate an important comment. It is true that when the meristem is multicellular, and the stem cell lines are retained during mutation accumulation (e.g. a structured meristem analyzed in Tomimoto and Satake 2023), it is possible for a mutation to have arisen earlier before the bifurcation. Using a mathematical model, we have proved that the intercept and slope of the linear regression between the pairwise genetic distance and physical distance are influenced by the type of a meristem (strength of somatic genetic drift in a meristem) as well as the branching architecture of the tree. We have included an explanation of this point in the revised manuscript (ll. 244–249).

1. Supplementary Data 7: Column J should be "2_2"

We corrected the typo.